# OpenReview forum: "CAMEx: Curvature-aware Merging of Experts"
_ICLR.cc/2025/Conference — ICLR 2025 Poster_

### Official Review · Reviewer_FRuk · 2024-10-30

**Soundness:** 3
**Presentation:** 3
**Contribution:** 3
**Rating:** 8
**Confidence:** 4

**Summary:**

The paper proposes a curvature-aware merging method using natural gradients. The experiments show that CAMEx can outperform Euclidean-based expert merging. I have read the paper several times and I really like the ideas from the paper. However, I would like to see more experimental result analysis about the model merging.

**Strengths:**

Curvature-aware merging is an interesting idea for expert merging.

**Weaknesses:**

1. In the section 3.2. "Ties-CA achieves the highest scores on SST-2 (94.61), MRPC (92.49), CoLA
(60.06), and MNLI (86.45), showing significant improvements over both the vanilla and standard
Ties models." Actually, the experiments in Table 2 don't show any significantance test. Could you present the significance test(T-test) of the improvement? Otherwise, you should reclaim these conclusions.

2. The experiment results are not strong enough to prove the effectiveness of CAMEx. Could you show some downstream zero-out results based on your T5 model? Please refer to https://arxiv.org/abs/2405.11157. For example, the downstream results on question answering (boolq, OpenbookQA) and reasoning (BBH),


3. The experiments only show the T5-base model. I am a little suspective that conclusions in the paper can also be generalized to other LMs. Could you show GLUE performance based on LLama-3 or phi-3( Vanilla, Ties, and Ties-CA is fine)?


I would like to raise my score if there is more experimental analysis.

**Questions:**

I may miss something. What are the training datasets for table 2?

---

> ### Author Response · Authors · 2024-11-19
> **Rebuttal (1/3): Weaknesses 1**
>
> **Weakness 1. In the section 3.2. "Ties-CA achieves the highest scores on SST-2 (94.61), MRPC (92.49), CoLA (60.06), and MNLI (86.45), showing significant improvements over both the vanilla and standard Ties models." Actually, the experiments in Table 2 don't show any significantance test. Could you present the significance test(T-test) of the improvement? Otherwise, you should reclaim these conclusions.**
>
> **Answer:** Thanks for your comments. Following your suggestion, we have conducted the following significance tests of the improvement.
>
> *T-test*: During the rebuttal period, we made every effort to conduct an additional experiment on a Sparse Mixture of Experts (SMoE) backbone with over 1 billion parameters. However, due to limited computational resources and the short timeframe, we were only able to provide results from seven different seeds for fine-tuning T5, T5-Ties, and T5-Ties-CA on SST-2, MRPC, CoLA, and MNLI. We report the t-test results, beginning with the null hypothesis:
>
> * $H_0$: The performance between each pair of T5-Ties-CA vs T5-Ties, and T5 on GLUE SST-2, MRPC, CoLA, and MNLI are the same.
>
> * We choose the significant value to be 0.05.
>
> We have provided the results for significance tests on SST-2, MRPC, CoLA, and MNLI benchmarks in Appendix G and included those results below.
>
>
> **SST-2:**
>
> Table 1. Evaluation results on SST-2 with different random seeds.
> | Index | Ties_CA | Ties  | Vanilla|
> |-------|---------|-------|-------|
> | 1     | 94.44   | 93.77 | 93.31 |
> | 2     | 94.86   | 94.13 | 93.33 |
> | 3     | 94.62   | 93.90 | 93.21 |
> | 4     | 94.60   | 94.12 | 93.46 |
> | 5     | 94.54   | 93.70 | 93.41 |
> | 6     | 94.55   | 93.87 | 93.56 |
> | 7     | 94.37   | 94.03 | 93.67 |
>
> Table 2. T-statistic and p-value when evaluating on SST-2.
> |Test | t-statistic | p-value |
> |-----|-------------|---------|
> |Ties-CA vs Vanilla | 13.72 | 1.08e-8 |
> |Ties-CA vs Ties    | 7.36 | 8.74e-6 |
>
>
> **MRPC:**
>
> Table 3. Evaluation results on MRPC with different random seeds.
>
> | Index | Ties_CA | Ties  | Vanilla|
> |-------|---------|-------|-------|
> | 1     | 92.35   | 91.35 | 89.85 |
> | 2     | 92.61   | 91.30 | 89.65 |
> | 3     | 92.55   | 91.40 | 89.74 |
> | 4     | 92.40   | 91.55 | 89.49 |
> | 5     | 92.54   | 91.62 | 89.76 |
> | 6     | 92.44   | 91.77 | 89.85 |
> | 7     | 92.33   | 91.43 | 89.62 |
>
> Table 4. T-statistic and p-value when evaluating on MRPC.
>
> |Test | t-statistic | p-value |
> |-----|-------------|---------|
> |Ties-CA vs Vanilla | 42.91 | 1.67e-14 |
> |Ties-CA vs Ties    | 12.95  | 2.06e-8 |
>
>
> **CoLA:**
>
> Table 5. Evaluation results on CoLA with different random seeds.
>
> | Index | Ties_CA | Ties  | Vanilla    |
> |-------|---------|-------|-------|
> | 1     | 61.01   | 57.95 | 57.74 |
> | 2     | 59.53   | 58.63 | 57.82 |
> | 3     | 60.36   | 58.90 | 58.03 |
> | 4     | 60.13   | 58.92 | 58.23 |
> | 5     | 59.41   | 58.31 | 58.51 |
> | 6     | 59.33   | 57.38 | 57.36 |
> | 7     | 60.03   | 58.53 | 58.40 |
>
> Table 6. T-statistic and p-value when evaluating on CoLA.
>
> |Test | t-statistic | p-value |
> |-----|-------------|---------|
> |Ties-CA vs Vanilla | 7.14 | 1.18e-5 |
> |Ties-CA vs Ties    | 5.16  | 2.00e-4 |
>
>
> **MNLI:**
>
> Table 7. Evaluation results on MNLI with different random seeds.
>
> | Index | Ties_CA | Ties  | Vanilla|
> |-------|---------|-------|-------|
> | 1     | 86.52   | 86.25 | 86.22 |
> | 2     | 86.45   | 86.32 | 86.31 |
> | 3     | 86.37   | 86.39 | 86.36 |
> | 4     | 86.59   | 86.46 | 86.41 |
> | 5     | 86.32   | 86.53 | 86.50 |
> | 6     | 86.54   | 86.38 | 86.34 |
> | 7     | 86.47   | 86.41 | 86.34 |
>
> Table 8. T-statistic and p-value when evaluating on MNLI.
>
> |Test | t-statistic | p-value |
> |-----|-------------|---------|
> |Ties-CA vs Vanilla | 2.29 | 0.04|
> |Ties-CA vs Ties    | 1.49 | 0.16 |
>
> Based on the p-values in the tables above, we draw the following conclusions:
> - The T5-Ties-CA variant significantly outperforms T5-Ties and T5-Vanilla on SST-2, MRPC, and CoLA.
> - While T5-Ties-CA does not statistically significantly outperform T5-Ties on MNLI, it still demonstrates significant improvement over T5-Vanilla. Consequently, we have revised our statement in the main text of the paper to reflect this clarification in line 386 of main text.

---

> ### Author Response · Authors · 2024-11-19
> **Rebuttal (2/3): Weakness 3**
>
> **Weakness 3. The experiments only show the T5-base model. I am a little suspective that conclusions in the paper can also be generalized to other LMs. Could you show GLUE performance based on LLama-3 or phi-3( Vanilla, Ties, and Ties-CA is fine)?**
>
> **Answer:** Thank you for your insightful suggestions. During the rebuttal, we made every effort to conduct additional experiments using Phi-3 with **3.8B** parameters and Phi-3 MoE with **7.4B** parameters on the GLUE benchmark. However, due to limited computational resources and the constrained rebuttal timeframe, the training processes remain incomplete. The currently available results are provided in the table below, and we will share the remaining results in the revised manuscript.
>
> Table 9. Performance of Phi-3-mini variants on the fine-tuning tasks for GLUE benchmark.
> |Model|Params| SST-2 | MRPC | CoLA  |STSB| RTE  |
> |-----|------|-------- | -------- |--------|--------|--------|
> |Phi-3 | 3.8B| 95.76    |90.12| 61.36  | 88.7| 80.51|
> |Phi3-Ties  |7.4B | 96.56 |92.25 |62.33|89.99 |87.73|
> |Phi3-Ties-CA |7.4B |**97.23** |**94.04** |**63.43**|**90.27** |**88.09** |
>
> Based on the results above, we draw the following conclusions for larger backbones:
> - The Ties-CA and Ties variants remarkably outperform the vanilla version, creating a substantial performance gap.
> - Ties-CA further enhances the performance of Ties across all listed tasks.
>
> Thus, we believe that curvature-awareness holds potential for improving other language models (LMs).

---

> > ### Author Response · Authors · 2024-11-22
> > **Update for Phi-3 MoE training**
> >
> > Dear Reviewer, we have continued training our model (Phi-3 MoE with 7.4B parameters) on the remaining GLUE tasks. However, some tasks, such as QQP, require significant time to train (e.g., over 2 days to complete a training job). We have partially updated the results here and will provide further updates as training progresses.
> >
> > |Model|QQP |
> > |-----| -------- |
> > |Phi-3         | 92.38  |
> > |Phi3-Ties-CA | **94.80** |

---

> ### Author Response · Authors · 2024-11-19
> **Rebuttal (3/3): Question 1**
>
> **Q1. I may miss something. What are the training datasets for table 2?**
>
> **Answer:** We, again, sincerely thank the reviewer for the thoughtful comment. In Table 2 in our main text, we fine-tune T5 variants separately for each of the GLUE tasks using only the training data for the corresponding task. We have added more details into the caption of Table 2 in the revised manuscript.

---

> ### Author Response · Authors · 2024-11-19
> **Rebuttal: Weakness 2**
>
> **Weakness 2. The experiment results are not strong enough to prove the effectiveness of CAMEx. Could you show some downstream zero-out results based on your T5 model? Please refer to https://arxiv.org/abs/2405.11157. For example, the downstream results on question answering (boolq, OpenbookQA) and reasoning (BBH).**
>
> **Answer:**  Thanks for your comments. We conducted experiments to evaluate its zero-shot performance on related tasks as suggested. Due to limited computational resources and the constraints of the rebuttal timeframe, we opted to use a smaller pre-trained dataset.
>
> Specifically, we fine-tuned our model on the HotpotQA dataset [1] and evaluated it in a zero-shot setting on QA tasks (BoolQ and OpenBookQA dataset) and reasoning task (BBH dataset).
>
> These experiments were designed to align with the experimental settings in the suggested work. For example, we employed token-length normalized scores to select continuations for evaluating multiple-choice tasks. The results are provided in Table 10 below for convenience.
>
> Table 10: Zero-shot results for T5 backbones.
> | Method    | BoolQ    | OpenBookQA    | BBH    |
> |-------------|-------------|-------------|-------------|
> | Vanilla| 40.92| 29.7| 11.43|
> | Ties| 40.86| 30.4| 12.41|
> | Ties-CA| **42.66**| **30.6**| **14.38** |
>
>
> We hope this addresses your concern and demonstrates the downstream effectiveness of CAMEx in these tasks.
>
> **Reference**
>
> [1] Yang, Zhilin, et al. "HotpotQA: A dataset for diverse, explainable multi-hop question answering." (EMNLP 2018).

---

> > ### Comment · Reviewer_FRuk · 2024-11-24
> >
> > Thanks for your experimental analysis. It is interesting that the HotpotQA dataset can help with the reasoning tasks. It would be good to do more zero-shot evaluations based on the Phi-3 or larger models in the next iteration (Especially for multi-task transfer learning, and catastrophic forgetting investigation). However, according to the limited time resources, I think the current version is good enough to be selected. I would like to raise the score.

---

> > > ### Author Response · Authors · 2024-11-24
> > > **Thanks for your endorsement!**
> > >
> > > Thanks for your response and additional suggestions. We appreciate your endorsement and will further explore zero-shot evaluations using the Phi-3 or larger models to investigate multi-task transfer learning and catastrophic forgetting in the next version of our manuscript.

---

> ### Author Response · Authors · 2024-11-22
> **Any Questions from Reviewer FRuk on Our Rebuttal?**
>
> We would like to thank the reviewer again for your thoughtful reviews and valuable feedback.
>
> We would appreciate it if you could let us know if our responses have addressed your concerns and whether you still have any other questions about our rebuttal. We would be happy to do any follow-up discussion or address any additional comments.
>
> If you agree that our responses to your reviews have addressed the concerns you listed, we kindly ask that you consider whether raising your score would more accurately reflect your updated evaluation of our paper. Thank you again for your time and thoughtful comments!

---

### Official Review · Reviewer_2ASa · 2024-11-04

**Soundness:** 3
**Presentation:** 4
**Contribution:** 3
**Rating:** 6
**Confidence:** 3

**Summary:**

The paper proposes CAMEx, a curvature-aware approach to merging experts in Sparse Mixture of Experts (SMoE) models. The work introduces a novel merging protocol that incorporates natural gradients to account for non-Euclidean parameter manifold geometry, a dynamic merging architecture that reduces computational costs while maintaining performance, and provides theoretical analysis showing how curvature-aware updates combine traditional domain-specific merging with task alignment adjustments. The empirical evaluation demonstrates consistent improvements across multiple NLP and vision tasks.

**Strengths:**

- The curvature-aware approach consistently outperforms traditional Euclidean-based merging methods across different tasks and architectures
- The dynamic merging architecture achieves the same number of experts but reduces FLOPs per token, providing a practical way to improve efficiency
- The Kronecker-based approximation for the curvature matrix is computationally practical and empirically effective based on ablation studies
- Strong theoretical analysis in Section 2.6 shows how gradients w.r.t curvature matrix combine domain-specific merging with an adjustment term for task loss alignment
- Thorough empirical evaluation across diverse tasks, including ablation studies examining impact of key parameters (α scaling factor, Kronecker rank)

**Weaknesses:**

- Technical aspects of the method need clearer explanation - the causal segmenting approach lacks sufficient background/motivation in the main paper, key equations for merging need step-by-step walkthrough, and curvature-based updates require more detailed explanation
- Performance improvements are somewhat modest, especially on the GLUE benchmark tasks
- Training for longer appears to reduce the performance gap between proposed methods and baselines (Figure 3)
- Impact on larger model scales is not thoroughly explored - unclear if improvements would be more pronounced with models bigger than GPT-2
- Missing analysis of how different token-to-expert routing strategies might affect the method's performance

**Questions:**

- How does the choice of token-to-expert routing method (token-choice vs expert-choice) affect CAMEx performance?
- Would the performance improvements be more significant when training larger models?

---

> ### Author Response · Authors · 2024-11-19
> **Rebuttal (1/3): Weakness 1 (1/2)**
>
> **Weakness 1. Technical aspects of the method need clearer explanation - the causal segmenting approach lacks sufficient background/motivation in the main paper, key equations for merging need step-by-step walkthrough, and curvature-based updates require more detailed explanation.**
>
> **Answer:** Thanks for your comments. We have provided the background of causal segmenting, the step-by-step walkthrough of the merging equations, and the detailed explanation of the curvature-based updates in Appendix F. of our manuscript. We have also included them below for reference.
>
> **1. Background of Causal Segmenting.** A significant advancement in SMoE design centers on fully differentiable architectures that eliminate the need for additional loss terms to stabilize training. In [1], a model was introduced that computes a weighted average of expert feed-forward networks (FFNs). For an input $x$ with corresponding routing weights, the output is defined as:
>
> $$
> o_x = \text{FFN}\left(h_x; \sum_{i=1}^{N} s_i \cdot \mathbf{E_i}\right), \quad \text{where} \quad s_i = \text{Softmax}({\bf{G}(h_x)})_i.
> $$
>
> However, applying this approach to autoregressive language models is computationally costly, as the merged FFN must be computed for each token in the sequence, leading to costs that scale linearly with the number of experts. An alternative based on pooling—routing via the sequence's average representation, as follows:
> $$
> s_i = \text{Softmax}\left(\bf{G}\left(\frac{\sum_{j=1}^L h_{x_j}}{L}\right)\right)\_i.
> $$
> This, however, disrupts the autoregressive property essential for pre-training. To address this, [2] introduced causal segment routing. This technique merges FFNs in an MoE layer by utilizing information from the preceding segment to process the current segment. Specifically, given a training instance $X$ consisting of $L$ tokens (e.g., $L = 4096$), we divide the instance into $N$ segments, each containing $T$ (e.g., $T = 256$) consecutive tokens. For the $k$-th segment $S_k$, where $k > 1$, we compute the average of the hidden representations from the previous segment $S_{k-1}$, denoted as $\bar{h}\_{k-1}$. By using the average hidden representation, the model can adapt to prompts of varying lengths during inference. The average hidden representation $\bar{h}\_{k-1}$ is then employed to determine the routing weights, leading to a merged expert $\bar{\bf{E}}$:
> $$
> \bar{h}\_{k-1} = \frac{1}{T} \sum_{x \in S_{k-1}} h_x, \quad s_i = \text{Softmax}(\bf{G}(\bar{h}\_{k-1})), \quad \bar{\bf{E}} = \sum_i s_i \cdot \bf{E}\_i.
> $$
> The merged expert $\bar{\mathbf{E}}$ is then used to process all tokens in the current segment $S_k$, i.e., $o_x = \text{FFN}(h_x; \bar{\mathbf{E}}), \forall x \in S_k$. This approach ensures that the model's routing decisions rely exclusively on data from preceding positions. For the first segment $S_1$, the segment's own representation is used to compute the merging weights for its FFN. To prevent information leakage, a stop-gradient operation is applied to $\bf{G}(\bar{h}\_1)$:
> $$
>     \bar{h}\_0 = \dfrac{1}{T}\sum_{x \in S_0}^T h_x
> $$
> These tokens are then used to calculate the scores for the merging procedure
>
> $$
> \begin{align}
> \tag{$\texttt{ROLLandDETACH}$}
>     s_0 &= \texttt{DETACH}\left(\bf{G}(\bar{h}\_1, k)\right) \\\\
>     s_i &= \bf{G}(\bar{h}\_{i-1}), \quad i=1,\dots, S-1
> \end{align}
> $$

---

> ### Author Response · Authors · 2024-11-19
> **Rebuttal (2/3): Weakness 1 (2/2)**
>
> **2. Key equation for merging of CAMEx.**
> $$
> \tag{CA-Merg}
>     \bf{\hat{E}}\_m^l = \bf{E}\_m^l + \alpha\sum_{i=1}^{N-1} \bf{M}\_i\cdot (s^l_i*\tau_i^l)
> $$
> In (CA-Merg) equation, we consider the merging of experts at $l$-th layer of the model. $\bf{E}\_m^l$ denotes the "base" expert that is not included in the routing process. $\tau^l_i = \bf{E}\_i^l - \bf{E}\_m^l$ denotes $i$-th the domain-vector that adapts the "base" expert to the corresponding domain. Finally, $s^l_i$ denotes the score of the $i$-th domain vector w.r.t the input. We view the merging of experts as a optimization problem where $\alpha*s^l_i$ acts as the adaptive learning rate. Therefore, it is straightforward to integrate natural gradient approach into this equation by introducing curvature matrices $\bf{M}\_i$. Due to the challenging tractability of the Fisher Matrix in the intermediate layers of deep models, we proposed to learn them empirically through backpropagation as indicated by eq. (6) in the main text and a similar approach using meta-learning [3].
>
> We have added information about motivations of causal segmenting in section 2.4 of main text and elaborated more on causal segmenting algorithm in appendix C.1.
>
> Key equation for merging in dynamic architecture:
>
>  \begin{cases}
> \tag{Dynamic-Merg}
>     \bf{E}\_m^{l+1} &= \bf{E}\_m^l + \dfrac{\alpha}{N-1}\displaystyle\sum_{i=1}^{N-1} \bf{M}\_i \cdot \tau_i^l \\\\
>     \bf{\hat{E}}\_m^{l+1} &= \bf{E}\_m^{l+1} + \displaystyle\alpha\sum_{i=1}^{N-1} \bf{M}\_i \cdot (s^{l+1}\_i*\tau_i^{l+1})
> \end{cases}
>
>
> The (Dynamic-Merg) system perform two steps which are calculating base expert for the next layer and perform (CA-Merge), respectively. For the first step, we eliminate the score and take the average of curvature-aware domain vector instead to avoid information leakage. The result then takes the role as the base expert for the next layer.
>
> **3. Curvature update.** We suppose the reviewer is referring to the gradient update of the curvature matrix. In the main text, we try to give an explanation of how our method we update the curvature matrix with the curvature information of the parameters space. To achieve that, we first take the derivative of equation (CA-Merge) w.r.t the curvature matrix $\bf{M}_i$:
> $$
>     \dfrac{\partial \bf{\hat{E}}\_m}{\partial \bf{M}\_j}
>     =  \left(\alpha s_j * \tau_j \right)
>     = \alpha s_j * (\bf{E}_j - \bf{E}\_m)
> $$
> To evaluate the gradient of the task loss $\mathcal{L}$ w.r.t $\bf{M}_i$ we apply the chain-rule:
> $$
> \begin{equation}
>     \dfrac{\partial \mathcal{L}}{\partial \bf{M}\_j}
>     = \dfrac{\partial \mathcal{L}}{\partial \bf{\hat{E}}\_m} \cdot \dfrac{\partial \bf{\hat{E}}\_m}{\partial \bf{M}\_j}
>     = \alpha s_j * \dfrac{\partial \mathcal{L}}{\partial \bf{\hat{E}}\_m} \cdot (\bf{E}\_j - \bf{E}\_m)
> \end{equation}
> $$
>
> **References**
>
> [1] Mohammed Muqeeth, Haokun Liu, and Colin Raffel. *Soft merging of experts with adaptive routing*. Transactions on Machine Learning Research, 2024. ISSN 2835-8856. URL https://openreview.net/forum?id=7I199lc54z.
>
> [2] Zexuan Zhong, Mengzhou Xia, Danqi Chen, and Mike Lewis. *Lory: Fully differentiable mixture-of-experts for autoregressive language model pre-training*. In First Conference on Language Modeling, 2024. URL https://openreview.net/forum?id=LKEJPySnlt.
>
> [3] Eunbyung Park and Junier B Oliva. *Meta-curvature*. (NeurIPS 2019).

---

> > ### Author Response · Authors · 2024-11-19
> > **Rebuttal (3/3): Weakness 2, 4 and Question 2**
> >
> > **Weakness 2. Performance improvements are somewhat modest, especially on the GLUE benchmark tasks**
> >
> > **Weakness 4. Impact on larger model scales is not thoroughly explored - unclear if improvements would be more pronounced with models bigger than GPT-2.**
> >
> > **Q2. Would the performance improvements be more significant when training larger models?**
> >
> > **Answer:** During the rebuttal, we have diligently conducted additional experiments on Phi-3 with **3.8B** parameters and Phi-3 MoE with **7.4B** parameters using the GLUE benchmark. However, due to resource limitations and the short timeframe, the training is still ongoing. The currently available results are presented in the table below, and the remaining results will be included in the revised manuscript.
> >
> > Table 1. Performance of Phi-3-mini variants on the fine-tuning tasks for GLUE benchmark.
> > |Model|Params| SST-2 | MRPC | CoLA  |STSB| RTE  |
> > |-----|------|-------- | -------- |--------|--------|--------|
> > |Phi-3 | 3.8B| 95.76    |90.12| 61.36  | 88.7| 80.51|
> > |Phi3-Ties  |7.4B | 96.56 |92.25 |62.33|89.99 |87.73|
> > |Phi3-Ties-CA |7.4B |**97.23** |**94.04** |**63.43**|**90.27** |**88.09** |
> >
> > From the results in the table above, we make the following observations:
> > - Ties-CA consistently outperforms both Ties and Vanilla variants on the GLUE benchmark, with the performance gap widening as the backbone size increases, e.g., with Phi-3.
> > - We believe this improvement is generalizable to other language models larger than GPT-2.

---

> ### Author Response · Authors · 2024-11-19
> **Rebuttal: Weakness 3, Weakness 5, Question 1**
>
> **Weakness 3. Training for longer appears to reduce the performance gap between proposed methods and baselines (Figure 3)**
>
> **Answer**: To address the concern that extended training might narrow the performance gap between our proposed methods and the baselines, we conducted additional experiments by training our model for longer iterations on the Wikitext-103 dataset. The performance gaps between methods remain stable starting around epoch 40. As shown in Appendix H.3, Figure 7 the trends demonstrate consistent improvements of our method over the baseline, with the gap remaining significant even after prolonged training.
> For convenience, we briefly present the final pre-training results (after 60 epochs) here:
>
> Table 2: Performance of GPT-2 small variants for
> the pre-training task on Wikitext-103.
> | Method | PPL |
> | -------- | -------- |
> | Vanila     | 22.28     |
> | Domain-specific     | 19.68     |
> | Domain-specific-CA (**Ours**)     | **17.39**     |
>
> **Weakness 5. Missing analysis of how different token-to-expert routing strategies might affect the method's performance.**
>
> **Q1. How does the choice of token-to-expert routing method (token-choice vs expert-choice) affect CAMEx performance?**
>
> **Answer:** We address both Weakness 5 and Question 1 at the same time. Here we demonstrate our merging approach with the following routing mechanisms:
>
> - Stable MoE routing [2]
> - Naive routing [4], the default routing method employed in our main paper.
>
> Note that the Curvature Aware model leverages the segment routing strategy (the causal segmenting strategy) proposed in Lory [1], enabling a direct comparison between our model and the expert choice method. We direct the reviewer to Table 18 in Appendix H.2 of our revised manuscript for the results, which are also presented below in Table 3 for convenience.
>
> Table 3: Performance of T5-base variants on the finetuning GLUE tasks
> | Method | MRPC| RTE | STSB | SST-2 |
> |----------|----------|----------|----------|----------|
> | Expert Choice MoE | **93.10**   | 66.78   | 89.19   | 93.80   |
> | Stable MoE routing CA |92.96 |**78.76** | **89.64** | **94.63**  |
> | Naive routing CA    | 92.49   | 78.70 | 89.56   | 94.61   |
>
> In Table 3, comparing CAMEx with the expert choice model, we find that our approach achieves higher performance. We attribute the remarkable gap in some tasks to the expert choice model's potential for token-dropping issues. In particular, at the segment level, this approach may drop entire segments, leading to a severe decline in performance. Notably, the Stable MoE routing CA variant outperforms the naive routing version, demonstrating that Curvature-Aware merging benefits from more advanced routing methods.  Therefore, Curvature Aware merging performance gain is orthogonal to other advanced routing mechanisms thus the two can benefit from the incorporation of each other.
>
> We are also currently working on experiments with other routing strategies, such as X-MoE routing [3], and will report the results in the next few days.
>
> **References**
>
> [1] Zhong, Zexuan, et al. "Lory: Fully Differentiable Mixture-of-Experts for Autoregressive Language Model Pre-training." (COLM 2024).
>
> [2] Dai, Damai, et al. "Stablemoe: Stable routing strategy for mixture of experts." (ACL 2022).
>
> [3] Chi, Zewen, et al. "On the representation collapse of sparse mixture of experts." (NeurIPS 2022).
>
> [4] Shazeer, Noam, et al. "Outrageously large neural networks: The sparsely-gated mixture-of-experts layer." (ICLR 2017).

---

> ### Author Response · Authors · 2024-11-22
> **Any Questions from Reviewer 2ASa on Our Rebuttal?**
>
> We would like to thank the reviewer again for your thoughtful reviews and valuable feedback.
>
> We would appreciate it if you could let us know if our responses have addressed your concerns and whether you still have any other questions about our rebuttal. We would be happy to do any follow-up discussion or address any additional comments.
>
> If you agree that our responses to your reviews have addressed the concerns you listed, we kindly ask that you consider whether raising your score would more accurately reflect your updated evaluation of our paper. Thank you again for your time and thoughtful comments!

---

> ### Author Response · Authors · 2024-11-25
> **Update on Question 1: More experimental results**
>
> Dear reviewer 2ASa,
>
> Building on our earlier response, we are pleased to share the latest findings from our experiments exploring additional routing strategies. These new results highlight the impact of X-MoE routing on CAMEx performance, offering detailed insights that complement our previous analysis. Additionally, we compare the baseline performance (i.e., the ties merging expert without Curvature Aware) under different routing mechanisms with the corresponding Curvature-Aware counterparts to see how different routing functions affect CAMEx performance. It is worth noting that Expert Choice routing is not compatible with the experts merging method, as discussed by Lory [2] in their Subsection 5.3. The findings are summarized in Table 4 below. The same table can be found in Table 18 in Appendix H.2 of our revised manuscript.
>
> Table 4: Performance of T5-base variants on the finetuning GLUE tasks
> | Method | MRPC| RTE | STSB | SST-2 |
> |----------|----------|----------|----------|----------|
> | Expert Choice MoE | 93.10   | 66.78   | 89.19   | 93.80   |
> | Stable MoE routing Ties  | 91.92 | 75.48  | 89.48   |93.37 |
> | **Stable MoE routing CA** |**92.96** |**78.76** |**89.64** | **94.63**  |
> | Naive routing Ties    | 91.44 | 75.54  |88.58   |93.92 |
> | **Naive routing CA**   | **92.49**  | **78.70** | **89.56**  | **94.61**   |
> |X-MoE routing Ties| 91.99 | 75.29  | 88.42   |93.26 |
> |**X-MoE routing CA**| **92.79** | **78.20**  |**89.26**   |**94.38** |
>
> **Conclusion**
> As can be seen in the above table, the results suggest that Curvature-Aware merging benefit from more advanced routing strategies. The CA model consistently outperforms the baseline with Ties merging across all routing mechanisms. Additionally, we observe that both Naive routing CA and X-MoE routing CA deliver robust performance across GLUE tasks, while Stable MoE routing CA emerges as the most reliable choice overall.

---

### Official Review · Reviewer_ukHP · 2024-11-06

**Soundness:** 3
**Presentation:** 3
**Contribution:** 2
**Rating:** 6
**Confidence:** 3

**Summary:**

The paper presents CAMEx (Curvature-Aware Merging of Experts), a novel method for merging experts in Sparse Mixture of Experts (SMoE) architectures. CAMEx leverages natural gradients to account for the non-Euclidean curvature of the parameter space, achieving more effective model alignment and improved performance.

**Strengths:**

1. Originality
CAMEx introduces a novel approach by leveraging natural gradients to accommodate the non-Euclidean geometry of the parameter space, moving beyond traditional Euclidean-based merging methods. This curvature-aware design aligns model updates more naturally with the parameter manifold’s structure, enhancing both optimization and generalization. Additionally, the dynamic merging architecture optimizes resource usage without performance loss, offering a new direction for efficiently scaling SMoE architectures. This geometry-aware expert merging approach represents a significant conceptual advance with potential applications in other areas of model optimization.
2. Quality
The paper is well-supported by both theoretical and empirical evidence, with robust experiments across multiple tasks demonstrating CAMEx’s effectiveness. The theoretical foundation reinforces the practical findings, adding to the reliability of the results.
3. Clarity
The paper is generally well-organized, presenting its methodology and results clearly. While some technical sections could be further optimized to enhance accessibility, the core contributions are conveyed effectively.
4. Significance
CAMEx addresses the critical need for scalable and efficient model architectures. Its consistent performance improvements across various tasks, coupled with reduced computational demands, underscore its practical impact. The success of CAMEx may also inspire further research into non-Euclidean methods within model optimization, highlighting its broader significance.
In summary, CAMEx’s originality, strong experimental support, and contribution to scalable architectures make it a technically sound and impactful addition to the field.

**Weaknesses:**

While CAMEx introduces a novel and promising curvature-aware approach, several areas could be improved to enhance the paper's rigor and impact.
1. Limited Baseline Comparisons
The paper primarily compares CAMEx with standard Euclidean-based and a few curvature-aware merging methods but lacks comparisons with a broader set of state-of-the-art SMoE techniques, especially recent adaptive or gradient-based approaches. Including a wider range of baselines would better contextualize CAMEx’s strengths and highlight its unique advantages.
Suggested Improvement: Expand comparisons to include additional recent methods in expert merging or routing within SMoE, particularly those emphasizing efficiency or scalability. This would reinforce CAMEx’s position within the current landscape.
2. Clarity in Theoretical Exposition
The theoretical basis for CAMEx, rooted in natural gradient theory and parameter manifold curvature, could benefit from added clarity. Specifically, the mathematical distinctions between CAMEx’s curvature-aware merging and Fisher-based methods may not be fully accessible to all readers, particularly regarding computational efficiency.
Suggested Improvement: Simplify or visually illustrate the theoretical explanation, potentially with diagrams or flowcharts, to clarify how CAMEx achieves computational advantages without sacrificing performance.
3. Limited Analysis on Hyperparameters
The paper provides only a brief discussion of key hyperparameters, such as the scaling factor α. Given that curvature-aware methods may be sensitive to parameter tuning, a more comprehensive analysis of these hyperparameters would offer valuable insights.
Suggested Improvement: Conduct a sensitivity analysis on α and other critical parameters, highlighting optimal settings across tasks. This would provide practical guidance for future users of CAMEx.

**Questions:**

1. Given CAMEx’s potential for scaling, have you tested its efficiency and performance on larger models? If such experiments were beyond the scope of this paper, could you provide any empirical or theoretical estimates of how CAMEx’s computational advantages scale with model size and expert count? This would better inform readers of its applicability to real-world, large-scale SMoE deployments.
2. Could you elaborate on the impact of key hyperparameters, particularly the scaling factor α, on CAMEx’s performance? A sensitivity analysis or guidance on optimal parameter settings across different tasks would be highly beneficial for practitioners looking to implement CAMEx effectively.

---

> ### Author Response · Authors · 2024-11-19
> **Rebuttal (1/2): Weakness 2**
>
> **Weakness 2. Clarity in Theoretical Exposition The theoretical basis for CAMEx, rooted in natural gradient theory and parameter manifold curvature, could benefit from added clarity. Specifically, the mathematical distinctions between CAMEx’s curvature-aware merging and Fisher-based methods may not be fully accessible to all readers, particularly regarding computational efficiency. Suggested Improvement: Simplify or visually illustrate the theoretical explanation, potentially with diagrams or flowcharts, to clarify how CAMEx achieves computational advantages without sacrificing performance.**
>
> **Answer:** As the reviewer suggested, we have added theoretical clarification and a figure showing the differences in computational pipeline between CAMEx Merging and Fisher Merging in Appendix A and Section 3.1 of the revised manuscript. Also, please allow us to explain the mathematical distinctions between these two methods to highlight how CAMEx achieves computational advantages without compromising performance.
>
> Both our CAMEx’s curvature-aware merging and Fisher-based methods aim to capture the curvature of the parameter space during the merging process. (Diagonal) Fisher Merging [1] applies a diagonal approximation to the Fisher information matrix. In particular, in [1], the authors estimate the diagonal of the Fisher matrix as:
> $$\bf{F} =\underset{x \sim D_m}{\mathbb{E}} \left[\underset{y \sim p_{\bf{\theta}}(y|x)}{\mathbb{E}} \left[ \nabla_{\bf{\theta}} \log p_{\bf{\theta}}(y|x)  \nabla_{\bf{\theta}} \log p_{\bf{\theta}}(y|x)^{\top} \right] \right].
> $$
> The expectation over $y$ can be estimated via sampling from $p_\theta(y|x_i)$ or computed exactly when the number of classes is small. The closed-form solution for Fisher merging (without necessarily applying the diagonal approximation) is given by:
> \begin{equation}
> \bf{\hat{E}}^l_m =   \bigg(\displaystyle \sum_{m=1}^M  F^l_m \bigg)^{-1} \bigg( \displaystyle\sum_{i=1}^N  F^l_i \bf{E}^l_i \bigg) .
> \end{equation}
> Thus, to approximate the Fisher Information Matrix for SMoE models, Fisher merging requires storing $\nabla_{E_i} \log p_{E_i}(y|x)$ for all experts and all $x$ in the training dataset. Additionally, it has been noted that Fisher merging can suffer from poor performance when fewer examples are used to estimate the Fisher matrix.
> Different from Fisher merging, by denoting $\mathbf{M_i}$ as the curvature matrix of the $i$-th expert, CAMEx utilizes the formula for merging experts derived from the natural gradient descent update as:
> \begin{equation}
>     \bf{\hat{E}}^l_m =\bf{E_m^l} + \alpha\sum_{i=1}^{N-1} \bf{M_i}\cdot (s^l_i*\tau_i^l).
> \end{equation}
>
> As can be seen in the equation above, CAMEx implicitly implements the gradient-based matching between the task loss gradient and domain-vector of the corresponding expert to approximate the empirical Fisher through the dynamic of gradient descend update of $\bf{M_i}$:
> \begin{align}
> \bf{M}^{t+1}_i &= \bf{M}^t_i - \beta * \dfrac{\partial \mathcal{L}}{\partial M^t_i} = \bf{M}^t_i - \alpha\beta * s^t_i * \dfrac{\partial \mathcal{L}}{\partial \bf{\hat{E}}^t_m} \cdot (\bf{E}^t_i - \bf{E}^t_m),
> \end{align}
> where the term $\dfrac{\partial \mathcal{L}}{\partial \bf{\hat{E}_m}} \cdot (\bf{E}_i - \bf{E}_m)$ represents the outer product of the gradients of the task loss and the domain vectors. This operation contributes to capturing the curvature of the expert parameter space, ensuring curvature awareness during the merging process. This approach eliminates the need to compute the inversion of the empirical Fisher Information Matrix, thereby reducing computational overhead while maintaining sensitivity to parameter space curvature.
>
> ### Reference:
> [1] Michael Matena and Colin Raffel. *Merging models with fisher-weighted averaging*. (NeurIPS 2022).

---

> ### Author Response · Authors · 2024-11-19
> **Rebuttal (2/2): Question 1**
>
> **Question 1. Given CAMEx’s potential for scaling, have you tested its efficiency and performance on larger models? If such experiments were beyond the scope of this paper, could you provide any empirical or theoretical estimates of how CAMEx’s computational advantages scale with model size and expert count? This would better inform readers of its applicability to real-world, large-scale SMoE deployments.**
>
> **Answer:** Thank you for your valuable feedback. Following your suggestion, we have conducted additional experiments on Phi-3 with **3.8B** parameters and Phi-3 MoE with **7.4B** parameters on the GLUE benchmark. However, due to limited resources and the tight timeframe, the training is still in progress. The results available so far are summarized in the table below, and the remaining results will be incorporated into the revised manuscript.
>
> Table 1. Performance of Phi-3-mini variants on the fine-tuning tasks for GLUE benchmark.
> |Model|Params| SST-2 | MRPC | CoLA  |STSB| RTE  |
> |-----|------|-------- | -------- |--------|--------|--------|
> |Phi-3 | 3.8B| 95.76    |90.12| 61.36  | 88.7| 80.51|
> |Phi-3-Ties  |7.4B | 96.56 |92.25 |62.33|89.99 |87.73|
> |Phi-3-Ties-CA |7.4B |**97.23** |**94.04** |**63.43**|**90.27** |**88.09** |
>
> As can be seen in the Table 1 above, CAMEx considerably outperforms both the vanilla Phi-3 and the Phi-3-Ties variants across GLUE tasks. Notably, the performance gap is even more pronounced compared to the results reported for T5 variants in Table 2 of the main text. This underscores the strong potential of CAMEx for scaling and its applicability to larger-scale language models.
>
> To provide theoretical insights into the computational advantages and scalability of CAMEx, we analyze three architectures: Densed, SMoE, and CAMEx. We focus solely on the computational cost within the feedforward layers. In the table below:
> - $W$ and $b$ represent the parameters of a feedforward layer as discussed in section 2 of [2].
> - $\hat{W}$ and $\hat{b}$ denote the parameters of the merged experts in CAMEx.
> - $\mathcal{G} = f(W_G, b_G)$ is the gating function used in both SMoE and CAMEx architectures.
>
> Table 2. Comparsion in computatinal cost of different architectures.
> | Architecture | Computational cost|
> |--------------|-------------------|
> |Densed model   | $O(\sigma(Wx+b))$ |
> |SMoE (top-$k$)|$k*O(\sigma(Wx+b)) + O(\mathcal{G})$|
> |CAMEx/ Merged SMoE |$O\left(\sigma(\hat{W}x+ \hat{b})\right) + O(\mathcal{G})$ |
> - The computational cost of CAMEx/Merged SMoE is comparable to that of the dense counterpart, as the dominant term in both cases is generally $O(\sigma(Wx+b))$.
>
> ### Reference
> [2] Shwai He, Run-Ze Fan, Liang Ding, Li Shen, Tianyi Zhou, and Dacheng Tao. *Merging experts into one: Improving computational efficiency of mixture of experts.* (EMNLP 2023)

---

> ### Author Response · Authors · 2024-11-19
> **Rebuttal: Weakness 1 (1/2)**
>
> **Weakness 1. Limited Baseline Comparisons The paper primarily compares CAMEx with standard Euclidean-based and a few curvature-aware merging methods but lacks comparisons with a broader set of state-of-the-art SMoE techniques, especially recent adaptive or gradient-based approaches. Including a wider range of baselines would better contextualize CAMEx’s strengths and highlight its unique advantages. Suggested Improvement: Expand comparisons to include additional recent methods in expert merging or routing within SMoE, particularly those emphasizing efficiency or scalability. This would reinforce CAMEx’s position within the current landscape.**
>
> **Answer:**  Thanks for your comments. Below we address your concerns by 1) integrating our CAMEx method with a recent method in expert merging and 2) validating the CAMEx method when using various routing strategies.
>
> **1. Integrating CAMEx with recent merging expert methods**
>
> To address the concern regarding baseline comparisons, we expanded our experiments to include a broader range of most recent merging expert methods. Specifically, we integrated our CAMEx method with the Twin-Merging approach (NeurIPS 2024) [4]. Key distinctions between CAMEx and Twin-Merging lie in their core mechanisms:
> - Our method is a non-Euclidean merging method, which utilizes the curvature-aware matrix, whereas Twin-Merging is a model merging method, which relies on Euclidean merging.
> - Our approach is specifically designed for finetuning, in contrast to Twin-Merging, which is intended for post-training.
> - Finally, our dynamic mechanism performs inter-layer to form the merged expert, unlike Twin-Merging, which uses within-layer pre-calculations for merging.
>
> To integrate our method with Twin-Merging, we first fine-tune the Curvature Aware model for a specific GLUE task. At test time, we apply the Twin-Merging algorithm to merge experts, referring to our approach as Twin-CA. Notably, we found Twin-Merging to be a simple yet powerful technique that is easy to implement and helps reduce memory usage during inference. We adhere to the original implementation settings, using a sparsity density value of 0.2. We refer the reviewer to Table 17 in Appendix H.1 of our revision for the results, and present them in Table 2 below as well for convenience.
>
> Table 2: Performance of Twin-Merging its Curvature Aware (CA) variant on GLUE tasks
> | Method   | MRPC    | RTE    | STSB    |
> |-------------|-------------|-------------|-------------|
> | Twin-Merging| 91.97| 72.20| 88.56|
> | Twin-CA (**Ours**)| **92.30**| **74.73**| **89.55**|
>
> The above results demonstrate the effectiveness of our CAMEx approach when integrated with the Twin-Merging mechanism on GLUE tasks, highlighting its strong potential for incorporation into more advanced merging techniques.

---

> ### Author Response · Authors · 2024-11-22
> **Rebuttal: Weakness 1 (2/2)**
>
> **2. Validating CAMEx when using various routing strategies**
>
> We also demonstrate our merging approach with the following routing mechanisms:
>
> - Stable MoE routing [6]
> - Naive routing [7], the default routing method employed in our main paper.
>
>
> Note that the Curvature Aware model leverages the segment routing strategy (the causal segmenting strategy) proposed in Lory [5], enabling a direct comparison between our model and the expert choice method. We direct the reviewer to Table 18 in Appendix H.2 of our revised manuscript for the results, which are also presented below in Table 3 for convenience.
>
>
> Table 3: Performance of T5-base variants on the finetuning GLUE tasks
> | Method | MRPC| RTE | STSB | SST-2 |
> |----------|----------|----------|----------|----------|
> | Expert Choice MoE | **93.10**   | 66.78   | 89.19   | 93.80   |
> | Stable MoE routing CA |92.96 |**78.76** | **89.64** | **94.63**  |
> | Naive routing CA    | 92.49   | 78.70 | 89.56   | 94.61   |
>
> In Table 3, comparing CAMEx with the expert choice model, we find that our approach achieves higher performance. We attribute the remarkable gap in some tasks to the expert choice model's potential for token-dropping issues. In particular, at the segment level, this approach may drop entire segments, leading to a severe decline in performance. Notably, the Stable MoE routing CA variant outperforms the naive routing version, demonstrating that Curvature-Aware merging benefits from more advanced routing methods.  Therefore, Curvature Aware merging performance gain is orthogonal to other advanced routing mechanisms thus the two can benefit from the incorporation of each other.
>
> **References**
>
> [3] Prateek Yadav, Derek Tam, Leshem Choshen, Colin Raffel, and Mohit Bansal. *Resolving interference when merging models*. (NeurIPS 2023)
>
> [4] Lu, Zhenyi, et al. *Twin-merging: Dynamic integration of modular expertise in model merging.* ( NeurIPS 2024).
>
> [5] Zhong, Zexuan, et al. *Lory: Fully Differentiable Mixture-of-Experts for Autoregressive Language Model Pre-training.* (COLM 2024).
>
> [6] Dai, Damai, et al. *Stablemoe: Stable routing strategy for mixture of experts.* (ACL 2022).
>
> [7] Shazeer, Noam, et al. *Outrageously large neural networks: The sparsely-gated mixture-of-experts layer.* (ICLR 2017).

---

> ### Author Response · Authors · 2024-11-22
> **Rebuttal: Weakness 3 & Question 2. (1/2)**
>
> **Weakness 3. Limited Analysis on Hyperparameters The paper provides only a brief discussion of key hyperparameters, such as the scaling factor α. Given that curvature-aware methods may be sensitive to parameter tuning, a more comprehensive analysis of these hyperparameters would offer valuable insights. Suggested Improvement: Conduct a sensitivity analysis on α and other critical parameters, highlighting optimal settings across tasks. This would provide practical guidance for future users of CAMEx.**
>
> **Question 2. Could you elaborate on the impact of key hyperparameters, particularly the scaling factor α, on CAMEx’s performance? A sensitivity analysis or guidance on optimal parameter settings across different tasks would be highly beneficial for practitioners looking to implement CAMEx effectively.**
>
> **Answer**:
>
> Thank you for your insightful comment. We partially analyze the key hyperparameters, including the scaling factor $\alpha$, in the ablation section of the main text. To provide further clarity and practical guidance, we conduct additional experiments to evaluate the sensitivity of CAMEx's performance to $\alpha$, as well as the effect of varying the number of experts.
>
> #### 1. Experiment on $\alpha$:
>
> We extend the range of $\alpha$ for the ablation study, specifically evaluating Dare-CA and Ties-CA with $\alpha \in [0.1, 1.6]$. The evaluation is conducted using 5 different seeds, and the results are averaged. We refer the reviewer to Figure 8 in Appendix H.4.1 of our revision for the results, and present them in Table 3, 4, and 5 below as well for convenience.
>
> Table 3: Test performance on MRPC with Curvature-Aware methods under varying $\alpha$ settings.
>
> |   | 0.1| 0.2|0.3|0.4|0.5 | 0.6|0.7|0.8|0.9|1.0|1.1|1.2|1.4|1.6|
> |--------|--------|--------|--------|--------|--------|-------|-------|-------|-------|-------|-------|-------|-----|-------|
> |Ties-CA|91.03|91.21 | 91.34|90.81| 91.47 | 91.91 | 92.13 | 91.53 | 91.09 | 92.95 | 91.89 | 91.92 | 91.50|91.03 |
> |Dare-CA|89.88|90.25|89.86| 89.74| 90.28 | 92.78 | 92.33 | 91.83 | 93.50 | 91.15 | 91.44 | 91.26 | 91.04|90.96 |
>
>
> Table 4: Test performance on RTE with Curvature-Aware methods under varying $\alpha$ settings.
>
> |   |0.1|0.2|0.3 |0.4 | 0.5 | 0.6|0.7|0.8|0.9|1.0|1.1|1.2|1.4|1.6|
> |--------|--------|--------|--------|--------|--------|-------|-------|-------|-------|-------|-------|-------|-----|-------|
> |Ties-CA| 63.54|66.78 |63.89 |68.98 | 71.77 | 73.62|72.85 | 73.23| 75.77 | 75.81 | 74.78|74.25| 72.50|71.36 |
> |Dare-CA|63.53|64.62 |66.78|68.52 | 68.92 | 73.69|72.54 | 75.46| 73.23 | 78.70 | 75.36|74.67| 73.65|70.59 |
>
> Table 5: Test performance on STSB with Curvature-Aware methods under varying $\alpha$ settings.
> |   |0.1|0.2 |0.3 |0.4 | 0.5 | 0.6|0.7|0.8|0.9|1.0|1.1|1.2|1.4|1.6|
> |--------|--------|--------|--------|--------|--------|-------|-------|-------|-------|-------|-------|-------|-----|-------|
> |Ties-CA|88.42 | 87.42|87.73 |88.74 | 88.70 | 89.44 | 89.85 | 89.59 | 89.93 | 89.54 | 88.19 | 87.93 | 87.99 |87.01 |
> |Dare-CA| 89.27|89.45 |89.38|88.95 | 89.09 | 89.42 | 89.49 | 89.75 | 89.38 | 89.56 | 88.74 | 87.71 | 88.09 |88.11 |
>
> The results in Tables 3, 4, and 5 above lead to the following observations:
>
>   - The performance of the models is suboptimal or even worse than the vanilla baseline when $\alpha$ is either too small ($\alpha \in [0.1, 0.4]$) or too large ($\alpha > 1.1$).
>   - **Dare-CA** is more sensitive to the choice of $\alpha$, showing sharper improvements and declines across the range.
>   - **Ties-CA** exhibits more gradual changes, suggesting it is more robust to variations in $\alpha$. The optimal range for $\alpha$ is $[0.8,1.0]$. As noted in the main paper, this observation aligns with the findings reported for the TIES model merging approach [3].

---

> ### Author Response · Authors · 2024-11-22
> **Rebuttal: Weakness 3 & Question 2. (2/2)**
>
> #### 2. Experiment on various number of experts:
> In addition to the experiments on various alpha values for GLUE tasks (i.e., MRPC, RTE, STSB) mentioned above, we conducted additional studies on our method using different numbers of experts in the T5 backbone. The impact of selecting the number of experts has been detailed in Figure 9, Appendix H.4.2 of our revised manuscript.
>
> Table 6: Performance of Curvature-Aware methods on MRPC with varying numbers of experts.
> |Num. Experts | 4| 5|6 |7| 8 |10  | 12 | 16  |
> |--------|--------|--------|--------|--------|--------|--------|--------|--------|
> |Ties-CA |89.48 |90.02 |92.33 | 92.40| 92.49|93.16 | 92.97| 93.72|
> |Dare-CA |88.99 |89.75 |91.93| 92.09 |92.28 |92.68 |92.96 |92.37 |
>
> Table 7: Performance of Curvature-Aware methods on RTE with varying numbers of experts.
> |Num. Experts |  4| 5|6 |7| 8 |10  | 12 | 16  |
> |--------|--------|--------|--------|--------|--------|--------|--------|--------|
> |Ties-CA |73.84 |74.11 |74.64 |75.20 | 75.81| 77.20|77.86 | 78.36|
> |Dare-CA |74.45 |75.20 | 76.50|77.36 |78.70  | 78.66|  78.85 |79.39 |
>
> Table 8: Performance of Curvature-Aware methods on STSB with varying numbers of experts.
> |Num. Experts |  4| 5|6 |7| 8 |10 | 12 | 16 |
> |--------|--------|--------|--------|--------|--------|--------|--------|--------|
> |Ties-CA |88.14 | 88.17 | 88.69|89.35 | 89.54| 89.33|89.41 | 89.84 |
> |Dare-CA | 88.97 |88.59 |89.27 |89.50 |89.56 |89.38  |89.92  |90.21 |
>
>
> The following conclusions can be drawn from Tables 6, 7, and 8 above:
>
> - Increasing the number of experts generally improves accuracy up to a certain point: Accuracy improves as the number of experts increases, with the most significant gains occurring from 4 to 8 experts.
> - After 12 experts, the accuracy either saturates or slightly decreases.
> - We suggest using 8 experts as it provides a balanced trade-off between performance and efficiency

---

> ### Author Response · Authors · 2024-11-22
> **Any Questions from Reviewer ukHP on Our Rebuttal?**
>
> We would like to thank the reviewer again for your thoughtful reviews and valuable feedback.
>
> We would appreciate it if you could let us know if our responses have addressed your concerns and whether you still have any other questions about our rebuttal. We would be happy to do any follow-up discussion or address any additional comments.
>
> If you agree that our responses to your reviews have addressed the concerns you listed, we kindly ask that you consider whether raising your score would more accurately reflect your updated evaluation of our paper. Thank you again for your time and thoughtful comments!

---

### Author Response · Authors · 2024-11-19
**Summary of Revisions**

Incorporating comments and suggestions from reviewers, as well as some further empirical studies we believe informative, we summarize here the main changes in the revised paper:
1. We have added theoretical clarification and a figure showing the different in pipeline between CAMEx merging and Fisher Merging in appendix A and section 3.1 of the revised manuscript.
2. We have added information about motivations of causal segmenting in section 2.4 of main text and elaborated more on causal segmenting algorithm in appendix C.1.
3. We reclaim our conclusion regardless of the significant improvement of Ties-CA base line to Vanilla and Ties baselines.
4. We have included additional experiments: integrating CAMEx into Twin-Merging (appendix H.1) and evaluating our merging approach with various routing mechanisms (appendix H.2).
5. We have added extended training graph for Wikitext-103 in appendix H.3.
6. We have conducted a more comprehensive ablation study on hyperparameters, including $\alpha$ and the number of experts, in appendix H.4.
7. Changes are made in blue in the revision.

---

### Author Response · Authors · 2024-11-19
**Global Response (1/2)**

Dear AC and reviewers,

Thanks for your thoughtful reviews and valuable comments, which have helped us improve the paper significantly. We are encouraged by the endorsements that: 1) The novelty of our approach using natural gradients to align with non-Euclidean parameter spaces, offering a novel, interesting and efficient scaling solution for Sparse Mixture of Experts (SMoE) architectures. (reviewers ukHP, FRuk); 2) The paper combines strong theoretical foundations with empirical evidence, demonstrating consistent performance improvements across multiple tasks. (reviewers ukHP, 2ASa); 3) The method addresses critical scalability challenges in model optimization and inspires further research into non-Euclidean methods, showcasing its significance and versatility. (reviewer ukHP)

One common concern raised by all reviewers is the performance of CAMEx on larger language models. We address this concern here and in each individual reviewer's comment.

---

> ### Author Response · Authors · 2024-11-19
> **Global Response (2/2)**
>
> During the rebuttal, we diligently conducted additional experiments on Phi-3 with **3.8B** parameters and Phi-3 MoE with **7.4B** parameters using the GLUE benchmark. However, due to resource limitations and the short timeframe, the training is still ongoing. The currently available results are presented in the table below, and the remaining results will be included in the revised manuscript.
>
> Table 1. Performance of Phi-3-mini variants on the fine-tuning tasks for GLUE benchmark.
> |Model|Params| SST-2 | MRPC | CoLA  |STSB| RTE  |
> |-----|------|-------- | -------- |--------|--------|--------|
> |Phi-3 | 3.8B| 95.76    |90.12| 61.36  | 88.7| 80.51|
> |Phi3-Ties  |7.4B | 96.56 |92.25 |62.33|89.99 |87.73|
> |Phi3-Ties-CA |7.4B |**97.23** |**94.04** |**63.43**|**90.27** |**88.09** |
>
> Our key observations are:
> - Performance Improvement with Scaling: Scaling up from Phi-3 (3.8B) to Phi3-Ties (7.4B) leads to notable gains in performance across all tasks, demonstrating the benefits of increasing parameter size.
> - Impact of Curvature-Aware Techniques: Phi3-Ties-CA, incorporating curvature-aware methods, achieves the highest scores across all tasks, underscoring the effectiveness of geometry-aware optimizations for improving model generalization and task-specific performance.
> - Broad Improvement Across Diverse Tasks: Phi3-Ties-CA shows consistent gains, particularly significant in MRPC (+1.79), CoLA (+1.10), and RTE (+0.36) compared to Phi3-Ties, highlighting its robust enhancements across varied linguistic challenges.
>
> We also address reviewer FRuk's concern on the significant improvement of CAMEx on some tasks on GLUE benchmark. We have conducted t-test following the reviewer's suggestion and draw two conclusions:
> - The T5-Ties-CA variant significantly outperforms T5-Ties and T5-Vanilla on SST-2, MRPC, and CoLA.
> - While T5-Ties-CA does not significantly outperform T5-Ties on MNLI, it still demonstrates significant improvement over T5-Vanilla. Consequently, we will revise our statement in the main text of the paper to reflect this clarification in line 366 of main text.
>
> --------
> We hope that our rebuttal has helped to clear concerns about our work. We are glad to answer any further questions you have on our submission and we would appreciate it if we could get your further feedback at your earliest convenience.

---

### Author Response · Authors · 2024-11-22
**Additional result on Routing Mechanisms and downstream Zero-shot tasks**

Dear Reviewers and Area Chairs,

We sincerely thank all reviewers for their thoughtful reviews and valuable feedback. In response, we have conducted additional experiments to clarify the impact of different routing mechanisms on the model’s performance which we updated in Appendix H.2 of the revised manuscript. Below, we summarize our findings:

**Table 1**: Performance of T5-base variants on the finetuning GLUE tasks
| Method | MRPC| RTE | STSB | SST-2 |
|----------|----------|----------|----------|----------|
| Expert Choice MoE | **93.10**   | 66.78   | 89.19   | 93.80   |
| Stable MoE routing CA |92.96 |**78.76** | **89.64** | **94.63**  |
| Naive routing CA    | 92.49   | 78.70 | 89.56   | 94.61   |

Comparing CAMEx with the expert-choice model, we observe that our approach achieves notably higher performance. We attribute the remarkable performance gap observed in some tasks to potential token-dropping issues inherent in the expert-choice routing strategy.  Additionally, the Stable MoE routing (CA variant) outperforms the naive routing version, highlighting that Curvature-Aware merging benefits from advanced routing methods. Importantly, the performance improvements from Curvature-Aware merging are orthogonal to those from other routing mechanisms, allowing the two approaches to complement and enhance each other effectively.

As suggested by Reviewer FRuk, we further demonstrate the effectiveness of CAMEx on downstream zero-shot tasks.

**Table 2**: Zero-shot results for T5 backbones.
| Method    | BoolQ    | OpenBookQA    | BBH    |
|-------------|-------------|-------------|-------------|
| Vanilla| 40.92| 29.7| 11.43|
| Ties| 40.86| 30.4| 12.41|
| Ties-CA| **42.66**| **30.6**| **14.38** |

Moreover, we have evaluated the performance of Phi-3-mini variants on fine-tuning tasks for the GLUE benchmark. While some results have been updated here, we will provide further updates as training progresses.

**Table 3**: Additional results of Phi-3 on GLUE task
|Model|QQP |
|-----| -------- |
|Phi-3         | 92.38  |
|Phi3-Ties-CA | **94.80** |

Once again, we deeply appreciate your constructive feedback and look forward to any additional comments.

---

### Meta-Review · Area_Chair_fjAc · 2024-12-19

**Metareview:**

This paper introduces CAMEx, a new approach that uses natural gradients to accommodate the non-Euclidean geometry of the parameter space, moving beyond traditional Euclidean-based merging methods. The paper is well-supported by both theoretical and empirical evidence, with robust experiments across multiple tasks demonstrating CAMEx's effectiveness. The paper is generally well-organized, presenting its methodology and results clearly.

The reviewers pointed out limited baseline comparisons,  limited analysis of hyperparameters and raised questions about scaling and real-world applicability. However, the authors actively addressed the reviewer concerns during the rebuttal period.

I recommend acceptance as a poster.

**Additional Comments On Reviewer Discussion:**

The authors addressed the reviewer comments by conducting additional experiments and providing further clarification.

The authors expanded their experiments to include a broader range of recent merging expert methods, specifically integrating CAMEx with the Twin-Merging approach. They also validated CAMEx using various routing strategies such as Stable MoE routing, Naive routing, and Expert Choice routing.

The authors added theoretical clarifications to better explain the mathematical distinctions between CAMEx's curvature-aware merging and Fisher-based methods.

The authors conducted additional experiments on larger models like Phi-3 with 3.8B parameters and Phi-3 MoE with 7.4B parameters on the GLUE benchmark to demonstrate CAMEx's potential for scaling and its applicability to larger-scale language models. They also provided theoretical insights into the computational advantages and scalability of CAMEx.

Overall, the authors actively addressed the reviewer concerns to strengthen their paper during the rebuttal period.

---

### Decision · Program_Chairs · 2025-01-22

Accept (Poster)